# Adipose-Derived Stem Cells: Angiogenetic Potential and Utility in Tissue Engineering

**DOI:** 10.3390/ijms25042356

**Published:** 2024-02-16

**Authors:** Felor Biniazan, Alina Stoian, Siba Haykal

**Affiliations:** 1Latner Thoracic Research Laboratories, Division of Thoracic Surgery, Toronto General Hospital Research Institute, University Health Network, 200 Elizabeth Street Suite 8N-869, Toronto, ON M5G2C4, Canada; felor.biniazan@uhn.ca (F.B.); alina.stoian@uhn.ca (A.S.); 2Division of Plastic and Reconstructive Surgery, Department of Surgery, University of Toronto, 200 Elizabeth Street Suite 8N-869, Toronto, ON M5G2C4, Canada

**Keywords:** adipose tissue, adipose-derived stem cells, angiogenesis, tissue engineering

## Abstract

Adipose tissue (AT) is a large and important energy storage organ as well as an endocrine organ with a critical role in many processes. Additionally, AT is an enormous and easily accessible source of multipotent cell types used in our day for all types of tissue regeneration. The ability of adipose-derived stem cells (ADSCs) to differentiate into other types of cells, such as endothelial cells (ECs), vascular smooth muscle cells, or cardiomyocytes, is used in tissue engineering in order to promote/stimulate the process of angiogenesis. Being a key for future successful clinical applications, functional vascular networks in engineered tissue are targeted by numerous in vivo and ex vivo studies. The article reviews the angiogenic potential of ADSCs and explores their capacity in the field of tissue engineering (TE).

## 1. Introduction

In the past three decades, TE and regenerative medicine have had a remarkable impact on in vivo and ex vivo research and innovation, driven by the quest for effective solutions to address various challenges in diagnosing and replacing diseased and injured tissues [1,2,3,4,5,6]. As the integration of cells into engineered tissues is crucial for achieving proper structure and function, cell-based therapies play an essential role in tissue formation and have notably evolved as innovative and routine clinical solutions worldwide. The self-renewal and differentiation potential of cells play a pivotal role in the repair and regeneration of new tissue under diverse conditions and aspects. These processes largely depend on factors such as cell types and sources, cell-seeding techniques, cell signaling, growth factors, scaffold-based approaches, bioreactors, vascularization strategies, etc. [5,7,8,9,10]. Finding the proper source of stem cells for TE is one of the most attractive prospects for biologists and bioengineers. The most common sources of stem cells include embryonic stem cells, which possess the ability to differentiate into cells of all three germ layers. Adult stem cells, such as hematopoietic stem cells, mesenchymal stem cells, and neural stem cells, demonstrate the capacity to differentiate into a limited range of cell types specific to the tissue of origin. Induced pluripotent stem cells offer the advantages of pluripotency and find extensive use in disease modeling, drug development, and potential therapeutic applications. Perinatal stem cells, including amniotic fluid stem cells, exhibit multipotent differentiation capabilities. Additionally, umbilical cord blood stem cells are employed in hematopoietic stem cell transplantation, while cord tissue stem cells (mesenchymal stem cells), dental pulp stem cells (multipotent stem cells), and ADSCs are also notable sources [11,12,13,14,15,16,17,18,19]. Several concerns have been raised regarding the procurement of human embryonic stem cells. It is debated how morally acceptable it is to use therapies based on early human embryo destruction. Besides ethical concerns, safety issues regarding the clinical use of human embryonic stem cells are discussed. The plasticity of these cells makes them difficult to control after in vivo transplantation. [20]. Compared to embryonic stem cells, autologous adult stem cells do not present any main immunologic or ethical problems [21]. Adult stem cells can be sourced from various tissues, such as blood, liver, muscles, the brain, bone marrow, and skin. The procurement processes for cells from these organs are both painful and expensive, and they are often associated with the risk of causing morbidity in the donor site tissue. For example, bone marrow aspiration may result in a low yield during processing, requiring an additional ex vivo expansion step to achieve an adequate number of cells for clinical application [22]. Even bone marrow has been deemed a basic source of stem cells; nevertheless, ADSCs may represent a cheaper, less invasive, and larger source of stem cells [23]. ADSCs have emerged as a promising candidate, captivating the attention of researchers and clinicians alike [9,24,25,26]. The unique properties of AT extend beyond their traditional role as energy storage depots. Notably, ADSCs exhibit angiogenic potential, contributing to the development of new blood vessels—a crucial process in tissue repair and regeneration [27,28,29]. This paper explores the multiple aspects of ADSCs, shedding light on their angiogenetic potential and elucidating their utility from the dynamic perspective of TE.

## 2. Adipose Tissue and Its Components

AT is a primary organ in the body, playing crucial roles in regulation and metabolism while also displaying significant regenerative potential [30]. This tissue serves as an energy reservoir for multiple organs and additionally participates in immune responses, thermogenesis, as well as the synthesis and release of various hormones and small molecules (Table 1) [31]. AT is classified into two main types—white adipose tissue (WAT) and brown adipose tissue (BAT)—and further categorized based on physiological location, including subcutaneous, visceral, epicardial, intramuscular, and intramedullary fat [32]. Both BAT and WAT possess functions related to the breakdown (lipolytic) and synthesis (lipogenic) of lipids. They play roles in the accumulation and dissipation of energy, respectively [33]. Adipocytes, the most common type of cell in AT, are dedicated to storing fat [34]. The adipocyte cell lineage originates from mesenchymal stem cells (MSCs), which differentiate into adipoblasts, and subsequently preadipocytes, signifying commitment to the lineage [35]. When a preadipocyte exits the cell cycle and starts accumulating fat deposits, it transforms into a fully matured adipocyte [31]. Interestingly, completely differentiated adipocytes exhibit characteristics of stem cells and have the potential for dedifferentiation. Approximately one-third of AT consists of fully differentiated adipocytes, while the remaining two-thirds comprise a large number of preadipocytes. This composition allows AT to maintain flexibility and respond to external stimuli in diverse ways [34]. The stromal vascular fraction (SVF) within AT consists of various cell types, including preadipocytes, fibroblasts, immune cells, and ECs [31]. ADSCs are present in SVF along with other cell types, including hematopoietic, endothelial and other cells [36]. In recent years, extensive research has been directed towards isolating and characterizing ADSCs from diverse AT sources in both animal and human models [37]. 

## 3. ADSCs Isolation, Proliferation and Differentiation Properties

The AT is made of a large number of adipocytes, connective tissue, vascular and neural tissues, and non-fat cells [47]. ADSCs have demonstrated the ability to differentiate into various cell lineages, including AT, bone, cartilage, cardiomyocytes, muscle, neuronal cells, and ECs (Figure 1) [48,49,50]. In 2013, Ogura et al. illustrated the presence of adipose [48,49,50] Multilineage Differentiating Stress Enduring (adipose-Muse) cells in adult human AT [51]. Additionally, it was reported that adipose-Muse cells are positive for the pluripotency markers SSEA3, TR-1-60, Oct3/4, Nanog, and Sox2. They can spontaneously differentiate into mesenchymal, endodermal, and ectodermal cell lineages with efficiencies of 23%, 20%, and 22%, respectively. Moreover, the use of specific differentiation media significantly enhances differentiation efficiency in adipose-Muse cells (82% for mesenchymal, 75% for endodermal, and 78% for ectodermal) [52]. These cells are capable of generating cell types representing all three germ layers from a single cell and effectively differentiating into targeted cells when induced by cytokines. There are reports indicating that ADSCs retain their phenotypic characteristics, differentiation capacity, and ability to proliferate even after undergoing 25 passages [53]. Moreover, in a comparative investigation involving stromal cells derived from bone marrow and AT obtained from identical donors, De Ugarte et al. illustrated that ADSCs achieved initial confluence within one week with approximately 5% of the cell number required for marrow cells. This indicates a greater proliferative capacity for ADSCs, suggesting that the heightened proliferative activity of the AT-derived population would expedite the generation of a clinically effective cell dose compared to bone marrow cells [54].

Additionally, the ADSCs yield is impacted by different factors, including AT location, age, species, and harvesting methods [55,56,57]. Subcutaneous depots, situated both superficially and deeply within the abdomen, are regarded as a superior reservoir of stem cells in comparison to other fat depots [58]. Additionally, a separate comparative study suggested that the stromal vascular fraction (SVF) obtained from superficial AT is better than other sources [59]. These collective findings highlight the potential of superficial abdominal AT as a promising source for ADSCs. Nevertheless, in contrast to the aforementioned studies, a recent investigation demonstrated a considerably higher yield of ADSCs and SVF in the inner and outer thigh regions when compared to those from the waist, abdominal, and inner knee regions [60]. AT is commonly acquired through two distinct methods: standard en bloc resection or lipoaspiration. The yield, viability, and growth attributes of ADSCs are notably influenced by the specific harvesting approach employed. Vermette et al. demonstrated a 1.8-fold higher cell yield at the point of extraction for lipoaspiration-derived cells, which exhibited similarly or slightly superior proliferation in culture compared to cells obtained through resection [61]. Nonetheless, studies indicate that ultrasound-assisted liposuction leads to a reduced frequency of proliferating ADSCs and an extended population doubling time when contrasted with the resection method [62]. In the process of harvesting ATs for the isolation of ADSCs, thorough evaluations have been conducted on factors such as donor site, gender, and age, which have also been widely assessed [63]. Research conducted by Buschmann et al. demonstrated a notable decrease in ADSC yield in the elderly population when compared to the middle-aged group [64]. In a rabbit model, aging was found to adversely affect ADSC yield and adipogenic potential, while osteogenic and clonogenic properties remained largely unaffected [65]. The systematic review published in 2017 provides evidence suggesting a decline in the proliferation and differentiation potential of ADSCs with advancing age and increasing body mass index [66]. Furthermore, evaluating the impact of donor health status on the viability and quality of ADSCs can provide valuable insights for optimizing cell-based therapies and regenerative medicine approaches in various clinical contexts. Shahram et al. analyzed the viability of ADSCs from healthy donors and gamma-irradiated ones. In conclusion, the authors suggested that gamma irradiation damages cells’ DNA and reduces cell viability [67]. A review of published articles on obesity and ADSCs from 2019 concluded that obesity reduces ADSC qualities and may affect the therapeutic value of ADSCs by reducing ADSC angiogenic differentiation, proliferation, migration, and viability [68]. The results of other studies have shown that diabetes can lower the activity of ADSCs in proliferation assays and alter their phenotypical characteristics. Type 2 diabetes has been found to affect the activity of stem cells, while insulin resistance influences the proliferation of ADSCs [69]. 

### 3.1. Impact on Angiogenesis

Being a key for future successful clinical applications, functional vascular networks are critical to ensuring viability, functionality and preventing poor outcomes. They are also dependent on the neovascularization potential of the graft [70,71,72,73]. The circulatory system plays a vital role in sustaining organ function by supplying oxygen and nutrients. While the turnover of the adult vasculature in many organs is generally low, the formation of new blood vessels occurs in response to ischemia, ensuring the delivery of oxygen to the ischemic tissues [74]. In the human body, depending on the tissue type, the average capillary density is approximately 40 µm, and typically metabolically active tissue is not more than 100–200 µm away from a neighboring vessel. This means that the smallest vessels (capillaries) play the main role in oxygen and nutrient diffusion [75,76]. Tissue repair is a blood-dependent process, and new blood-vessel formation occurs through angiogenesis and neovasculogenesis [77,78,79]. While both processes share the common objective of generating new blood vessels, angiogenesis specifically involves the creation of new branches or protrusions from pre-existing vessels. In contrast, neovascularization pertains to the development of entirely new blood vessels [31]. Angiogenesis is a process governed by growth factors and cellular signaling pathways. Comprehending the molecular mechanisms underlying angiogenesis is crucial. Sprouting angiogenesis for vessel development commences with the release of proangiogenic growth factors along a gradient from oxygen- and nutrient-deprived microenvironments. These factors prompt quiescent ECs to transition into an activated state. Among the key underlying mechanisms, activated ECs produce matrix metalloproteinases (MMPs) to degrade the basement membrane, enabling them to exhibit invasive properties. Consequently, endothelial tip cells extend protrusions and migrate toward the origin of the angiogenic stimulus. Additionally, endothelial sprouts are believed to possess filopodia at their tips, facilitating cell migration and guidance [80]. After the guidance provided by tip cells, stalk cells undergo proliferation to facilitate the elongation of the sprouting process and may establish lumens. Tip cells from adjacent sprouts merge to form vessel loops, known as anastomosis, contributing to the extension of the lumen and the initiation of blood flow within the newly formed vessels. In research conducted by Gerhardt et al., it was observed that endothelial sprouts elongate filopodia in reaction to extracellular VEGF-A signaling [81,82]. VEGFR2 plays a role in controlling cell migration. VEGFR2, along with its ligand VEGFA, plays a crucial role in neovascularization. Additionally, vascular stalk cells were stimulated by VEGF-A to proliferate upon the binding of the growth factor to its receptor [83]. Stalk cell proliferation and paracrine factors, like granulocyte-colony stimulating factor (G-CSF), which aids in recruiting stem cells, contribute to the formation of these structures. Chemotactic factors, such as the chemokines stromal cell-derived factor 1 (SDF-1) and VEGF, have demonstrated attractive properties and may serve to attract progenitor cells to ischemic sites [22]. Additionally, chemokines like MCP-1 or IL-8, which are associated with immune-competent cells, represent another example of agents that can attract progenitor cells [84]. The mechanism by which ADSCs manage tissue repair and regeneration is mainly accomplished by the secretion of growth factors (FGF, HGF, PDGF), the release of immunomodulatory cytokines, and the ability to differentiate into different cell types [26]. The neoangiogenesis mechanism might be both the direct differentiation of ADSCs into ECs and the indirect effect of angiogenic GFs released from ADSCs [85]. VEGF stands out among these growth factors as one of the most potent regulators of vascular biology. It plays a pivotal role in maintaining vascular integrity and stimulating both angiogenesis and vasculogenesis [86]. Another significant growth factor is fibroblast growth factor 2 (FGFB), known for its pro-angiogenic effects by contributing to extracellular matrix degradation [87]. Platelet-derived growth factor (PDGF) serves as a key mitogen for ECs, particularly at sites of vascular injury, and is released by platelets [88]. Transforming growth factor β (TGF-β) is essential for normal vascular development and the formation of tube-like structures [89]. Additionally, hepatocyte growth factor (HGF) is also noteworthy for its ability to enhance endothelial cell proliferation and migration [90]. ADSCs can contribute to angiogenesis by differentiating into ECs. This differentiation process is facilitated by the use of endothelial cell growth medium supplemented with growth factors and proteins such as VEGF and IGF-1, which are essential for promoting endothelial differentiation [91]. Research conducted in an ischemic model has shown that a three-dimensional (3D) culture, relying on hADSCs’ adhesion to a substrate, augments the endothelial differentiation of hADSCs. This enhancement is characterized by heightened expression levels of angiogenesis-promoting proteins, including matrix metalloproteinase (MMP)-9, HGF, VEGF, and IL-8 [92]. FGFB, another critical factor in the differentiation of hADSCs into ECs, functions by activating the protein kinase B (AKT) pathway and phosphorylating FOXO1 (forkhead box protein O1) [93,94]. Moreover, hypoxic conditions applied during cell culture can enhance the endothelial differentiation of hADSCs by upregulating the expression of paracrine factors such as VEGF [95]. Studies propose that ischemic conditions induce plaque angiogenesis, resulting in significantly larger plaque sizes following the regeneration of endothelial cell populations [22]. Various methods exist for augmenting the angiogenesis induced by ADSCs. One of the limitations of stem cell therapy strategies with ADSCs, especially conventional injection-based cell delivery, is that the procedure’s efficacy is significantly hindered by the low engraftment rate, primarily due to cell losses during transfer and subsequent cell death after initial retention. The implantation of ADSCs in scaffolds has the potential to overcome the limitations of stem cell therapy. Using scaffolds can ensure cellular cohesion and facilitate the co-delivery of ADSCs [96]. Wang et al. examine the function of ADSCs preloaded with superparamagnetic iron oxide (SPIO) nanoparticles in enhancing heart function. It demonstrates that SPIOASCs can differentiate into ECs, promote angiogenesis, and suppress ischemic cardiomyocyte apoptosis [97]. Delivery methods capable of transporting progenitor ECs or growth factors, like heparin-pluronic (HP) nanogels, could be employed to promote angiogenesis [98]. Moreover, a precise balance of growth factors is essential for orchestrating effective angiogenesis. On the other hand, more than 30 miRNAs have been identified to either upregulate or downregulate angiogenesis, and mRNAs can exert downstream effects influencing angiogenesis, such as enhancing the expression of VEGF. Specifically, miR-126 and miR-132 are two well-studied miRNAs intricately involved in promoting the proliferation and migration of ECs during blood vessel formation [31,99]. ADSCs also have the ability to release diverse growth factors and microRNAs during the process of neovascularization. In hypoxic environments, ADSCs exhibit a significant increase in the secretion of VEGF, a crucial angiogenic factor. Transplantation of ADSCs preconditioned with VEGF led to enhanced capillary density and the restoration of cardiac function in the ischemic myocardium, consequently resulting in a substantial improvement in maximal collateral blood flow delivery [100]. Additionally, ADSCs release microvesicles (MVs)/exosomes, which are enriched with miRNAs. MicroRNAs (miRNAs) derived from ADSCs are integral to the neovascularization process by stimulating the proliferation and migration of ECs. Kang et al. illustrated the involvement of the miR-3/hypoxia-inducible factor-1 (HIF-1) pathway in facilitating the migration and formation of tubes in human umbilical vein endothelial cells (HUVECs), as well as in the outgrowth of microvessels in mouse aortic rings and the development of vasculature in mouse Matrigel plugs [101]. Moreover, Togliatto et al. documented that the diminished expression of miR-126 in ADSCs from obese individuals suppressed the extracellular signal-regulated protein kinase 1/2/mitogen-activated protein kinase (Erk1/2/MAPK) pathway within the cells, resulting in compromised angiogenesis. They further suggested that the upregulation of miR-126 could effectively restore the angiogenic capacity of ADSCs in obese individuals [102]. The MV/exosome method, when contrasted with the stem cell therapy approach, mitigates the risk of tumor formation resulting from unregulated cell proliferation and microvasculature blockage upon the intra-arterial introduction of implanted cells [101]. Treatment of ADSCs with various pharmacological agents, such as rosuvastatin, ghrelin, etc., has enhanced their angiogenesis ability and paracrine function. Zhang et al. examined whether the administration of rosuvastatin enhanced the survival of ADSCs following transplantation into infarcted hearts and indicated that rosuvastatin could potentially enhance the viability, paracrine activity, and angiogenesis ability of transplanted ADSCs, possibly by activating the PI3K/Akt and MEK/ERK1/2 signaling pathways in a rat myocardial infarction model [103]. Han et al. also reported that ghrelin could emerge as a viable option for hormone-driven approaches aimed at enhancing the effectiveness of ADSCs for cardiac ischemic diseases through the PI3K/AKT pathway [104]. Further research is warranted to elaborate on the mechanisms of these effects. 

### 3.2. Endothelial Cells Differentiation

It is feasible to isolate and procure significant quantities of ECs from AT [105,106]. The purified ECs from both adipose and dermal origins exhibited comparable expression levels of standard endothelial surface markers, including PECAM-1, VE-Cadherin, and VEGFR2 [107]. The observation that adipose-derived ECs exhibit enhanced proliferation under poor conditions compared to dermal-derived ECs suggests a potential for superior proliferation when used in wound treatment under poor conditions [107]. Despite the naturally differentiated ECs, considerable attention is devoted to generating ECs from resident stem and progenitor cells in AT [6]. The differentiation capacity of ADSCs into ECs was initially demonstrated in 2004. In this study, they evaluated the effectiveness of injecting cultured human SVF cells to promote revascularization in hindlimb ischemia in immunodeficient nude mice after 15 days. The results demonstrated that these cells markedly enhanced the angiographic score and cutaneous blood flow compared to untreated hindlimb ischemic nude mice. Evaluation of the potential of SVF-derived cells to integrate into newly formed blood vessels showed numerous human CD31-positive cells in mice hindlimbs injected with SVF. In comparison, no CD31-positive cells were observed in the non-injected hindlimbs. These findings strongly indicate that, in vivo, SVF-derived cells of human origin can differentiate into ECs and actively participate in vascular regeneration. They also evaluated the in vitro endothelial differentiation capacity of cultured SVF cells. Their results showed that cultured human SVF cells in an endothelial differentiation medium, after 10 days, formed branched tubular structures, with the majority showing strong positivity for CD31 and vWF, both of which are endothelial markers. Additionally, dedifferentiated mature human adipocytes exhibit the potential to acquire the endothelial phenotype in vitro and promote the formation of vessel-like tubes, which may suggest a potential common precursor for cells of endothelial and adipocyte phenotypes [108]. Konno et al. showed that the absence of bFGF significantly reduced the capacity of ADSCs to uptake Ac-LDL and resulted in the downregulation of EC marker expression, which highlighted bFGF as a highly effective inducer of EC differentiation. The CD34+/CD31– subpopulation demonstrated the ability to differentiate into ECs after culturing in supplemented endothelial growth medium with IGF and VEGF. Under these conditions, the cells exhibited a spindle-shaped morphology and displayed elevated expression levels of EC markers, including vWF and CD31 [109]. In the in vitro study, Cao et al. isolated a distinct cell subset of ADSCs identified as (CD34–/CD31–) and cultivated them on matrigel supplemented with bFGF and VEGF. The features of this subset were aligned with those of human umbilical vein ECs. In their in vivo study, they demonstrated that ADSCs have the ability to undergo differentiation in response to local signals, giving rise to ECs that actively participate in neoangiogenesis in hindlimb ischemia models [110]. Fan et al. illustrated that ADSCs exhibit faster differentiation into ECs and display a more robust proliferation capacity compared to bone marrow-derived mesenchymal stem cells (BM-MSCs) [111]. 

MSCs derived from AT exhibit the capability to enhance neovascularization by directly differentiating into ECs. Remarkably, the support of hematopoiesis by adipose-derived MSCs is more potent both in vivo and in vitro compared to MSCs derived from bone marrow. Crucially, while there is evidence demonstrating the differentiation of MSCs into ECs, it is probable that, upon implantation at the site of injury, MSCs predominantly contribute to angiogenesis indirectly. This indirect involvement occurs through the secretion of factors that stimulate the differentiation of resident cells and promote angiogenesis [31]. Santerre et al. indicated that the elevation of EC markers in porcine ADSCs induced by VEGFR2 is facilitated through the activation of the ERK signaling pathway [112]. On the other hand, Santerre et al. highlighted that human adipose tissue-derived microvascular endothelial cells (HAMVECs) demonstrate characteristic features in terms of morphology, molecular profile, and functional attributes typical of ECs. However, the present study advocates for the application of HAMVECs to the endothelialization and vascularization of engineered tissues. The study emphasizes that ADSCs seem particularly valuable for vascular TEdue to their ability to remodel the extracellular matrix and serve as mural cells. Their findings indicate that ADSCs may contribute to the stabilization and maturation of the endothelium, partly by aiding in the assembly of its basement membrane [6]. 

Endothelial progenitor cells (EPCs) represent a crucial category of stem cells with robust angiogenic potential, playing a pivotal role in angiogenesis and vasculogenesis [113]. EPCs have proven effective in treating stroke, hindlimb ischemia, myocardial infarction, and diabetic foot ulcers [114,115,116,117]. However, the primary challenge associated with EPC-based transplantation lies in the limited quantity of these cells [118]. EPCs, sourced from umbilical cord blood, bone marrow, or peripheral blood, are scarce. To overcome this limitation, Van Pham et al. propose AT as a novel source of EPCs. In their investigation, EPCs derived from AT exhibited the expression of EC markers, including CD31 and VEGFR2, and demonstrated in vitro blood vessel formation similar to HUVECs. The study underscores the feasibility of isolating EPCs from AT by selecting slowly adherent cells from SVFs. These findings highlight the potential and rationale for utilizing AT in EPC isolation for the treatment of vascular diseases [118]. It is crucial to note that EPCs derived from AT exhibit greater proliferative capacity compared to EPCs sourced directly from ADSCs [119].

### 3.3. Regenerative Features and Utility in Tissue Engineering

ASCs have earned significant attention in the field of TE. Making a summary of ADSCs features, this type of cells can be categorized as low immunogenicity cells, multipotent, self-renewal, immunomodulatory, with paracrine effects, and of course, angio progenitor cells [120,121]. Their regenerative features, appropriate even for cadaveric ADSCs, make them promising candidates for developing novel therapeutic strategies to treat various diseases and injuries [122,123]. Highlighting the versatility of ADSCs and demonstrating their potential across ex vivo and in vivo applications, ADSCs have shown great utility in all TE fields: 

Adipose TE: In 2006, Matsumoto et al. first reported a technique named cell-assisted lipotransfer, combining aspirated fat with ADSCs. This process converts ADSCs-poor aspirated fat to ADSCs-rich fat, improving the efficacy of adipose transfer through transferred fat survival. ADSCs-rich fat helps restore tissue vascularization and organ function [124]. Since then, ADSCs have been used to engineer AT and improve outcomes in fat grafting for tissue augmentation [125,126,127,128]. Currently, ADSCs are widely used in breast reconstruction, especially for breast cancer patients [129,130]. 

Bone TE: The ability of ADSCs to differentiate into osteoblasts and to produce various growth factors and cytokines that stimulate the proliferation and differentiation of osteoprogenitor cells is used in TE to regenerate bone tissue in cases of bone defects, fractures, nonunion or delayed union [131,132,133,134]. ADSCs have also been shown to promote angiogenesis, which is crucial for supplying nutrients and oxygen to the healing or degenerative bone, or ADSCs may enhance the activity of bone marrow stem cells, encouraging in this way bone regeneration [135,136,137]. Even more, ADSCs can be combined with scaffold materials to promote bone growth and reduce immune responses in allotransplantation [138,139].

Cartilage TE: ADSCs have been investigated for their potential to regenerate cartilage due to their differentiation capacity into chondrocytes and the properties of ADSCs-derived exosomes to mitigate chondrocyte degradation [140]. ADSCs have recently been studied for the treatment of knee osteoarthritis. ADSC injections showed improvement in cartilage integrity [141,142]. Also, ADSCs may be an attractive therapeutic option for patients with rheumatoid arthritis due to the immunomodulatory abilities of these cells [143].

Muscle TE: ADSCs have been explored for applications in treating muscle injuries and degenerative muscle disorders, restoring muscle function and stimulating local angiogenesis [144]. The capacity of modified ADSC cells to contribute to muscle repair and their potential to deliver a repairing gene to dystrophic muscles have been highlighted by some in vivo studies [145,146]. Volumetric muscle loss can also be repaired by ADSCs combined with tissue-engineered constructs [147,148]. 

Corneal TE: Exploring an effective treatment that could achieve multi-dimensional repair of the injured cornea, the use of ADSCs was reported as an important component in the healing of corneal epithelium and limbus, the inhibition of corneal stromal fibrosis, angiogenesis, lymphangiogenesis, and also in the repair of corneal nerves, including diabetic corneal epithelial healing [149,150,151,152].

Nerve TE: ADSCs have been investigated for their potential in nerve regeneration; studies show the ability of ADSCs to repair acute sciatic nerve injury in rabbits [153]. They may be used in combination with nerve guidance conduits or other biomaterials to enhance nerve repair after injury or can be used to increase neural marker expression in an environment similar to the central nervous system [154].

Liver and Lungs TE: ADSCs can differentiate into various cell types, including hepatic and lung epithelial cells, showing promising application in TE for treatment or the development of bioartificial constructs [155,156]. Exosomes derived from ADSCs can suppress the progression of liver fibrosis [157,158]. ADSCs therapy minimized lung damage after ischemia-reperfusion injury in a rodent model by suppressing oxidative stress and inflammatory reaction. Also, administration of ADSCs (in combination with other medications) is a beneficial therapeutic approach in lung transplantation for rejection prevention [159,160].

Tissue healing: ADSCs can accelerate wound healing and/or improve healing by promoting angiogenesis, particularly in chronic wounds [154,161]. Their regenerative and immunomodulatory properties contribute to cell proliferation and accelerate wound healing and cutaneous regeneration [162,163,164,165].

Cardiac TE: ADSCs have been studied for their potential to repair damaged heart tissue after a myocardial infarction. They may contribute to the regeneration of cardiac muscle and improve heart function [166,167,168]. The studies also unveiled the healing potential of ADSC-derived extracellular vesicles and sub-populations of regenerative ADSCs, promising novel opportunities for improved cardiac healing following ischemic injury [169]. ADSCs can differentiate into cardiomyocyte-like cells and protect pre-existing cardiac cells through their paracrine activity, releasing antiapoptotic factors. Some studies present ADSCs induced with TGF-β1 as a good choice for stem cell therapy in cardiovascular diseases. [170,171]. Cardiomyogenesis is an extremely complex process that depends on the different signaling pathways. Cardiomyocyte development in vitro has a high degree of complexity, and it is not yet known the exact protocol for in vitro cardiac differentiation of the ADSCs [172]. 

Vascular TE: ADSCs play one of the central roles in promoting the formation of new blood vessels. This property is the most valuable in engineering vascularized tissues and improving the blood supply to implanted tissues, and for this compartment, all the characteristics and properties of ADSCs described previously can be repeated. It is relatively difficult for ADSCs to differentiate into vascular ECs and interconnect in the vascular network [173]. However, when ADSCs are co-cultured with different specific endothelial cells (HUVECs, human cardiac tissue ECs, and human pulmonary artery ECs) and cultivated in specific conditions (hydrogel construct, ECs medium, supplements like epidermal growth factor, vascular endothelial growth factor, insulin-like growth factor 1, bFGF, FBS, and antibiotics), the cells successfully differentiate into vascular ECs or can develop into a stable vascular network [174]. Several other ex vivo and in vivo studies have shown the beneficial effects of ADSCs in TE (Table 2).

The differentiation potential of ADSCs can be regulated by various molecular pathways. In vitro differentiation of MSCs can be set up by using a combination of differentiation growth factors and molecules. For example, in osteoblast differentiation, more BMP4 and BMP7 that belong to the transforming growth factor-beta (TGF-β) superfamily [198], miR-1 could promote the ADSCs’ differentiation into cardiomyocyte-like cells and express cardiomyocyte-specific markers in the myocardial microenvironment [199]. Stromal cells isolated from AT expanded and grown with chondrogenic media in alginate culture present synthesis of the cartilage matrix molecules including collagen type II, VI, as well as chondroitin 4-sulfate, also basal medium with insulin, transferrin, and selenious acid (ITS+) combined with TGF-beta1-stimulated ADSCs chondrogenic differentiation. treated with different doses of dexamethasone [200,201]. Transdifferentiated ADSCs showed positive expression of corneal epithelial marker CK3/12 on immunostaining [202], and ADSC differentiation into Schwann cell-like cells (dhASCs) using specific medium and glial growth factors is described [203].

In general, ADSCs regenerative capabilities can be summarized in a few important mechanisms: 

Immunomodulatory properties—ADSCs exhibit immunomodulatory properties by secreting factors that suppress inflammation and modulate immune responses. They can regulate the activity of immune cells such as T cells, B cells, dendritic cells, and macrophages through mechanisms involving cytokines like IL-10, TGF-β, Indoleamine-2,3-dioxygenase, and PGE2 (Prostaglandin E2) [204].

The anti-inflammatory properties—ADSCs are considered powerful suppressors of immune response, inhibitors of inflammatory cytokines (IFN-γ, TNF-α, and IL-12), and down-regulators of Th1-type cytokine expression. ADSCs suppress T cell allo-proliferation, secrete a high amount of immune suppressive cytokines, such as IL-6 and transforming growth factor-β1 (TGF-β1), and increase the number of CD4 T cells producing IL-10 [205].

Angiogenic and hematopoietic properties—the cytokine profile of ADSCs involved in angiogenesis and hematopoiesis includes interleukins (IL-6, IL-7, IL-10, IL-11), vascular endothelial growth factor (VEGF), basic fibroblastic growth factor (bFGF), tumor necrosis factor-alpha (TNF-α), granulocyte colony-stimulating factor (G-CSF), and macrophage colony-stimulating factor (M-CSF) [206].

ECM remodeling—rapid and quality remodeling of the ECM after fat grafting can happen by promoting neovascularization, regulating stem cell differentiation, and suppressing chronic inflammation [207]. 

In conclusion to this section, we can certainly put into discussion the factors that contributed to the in vivo and in vitro differentiation of ADSCs. The tissue culture microenvironment, cell passage numbers, cell source, donor variability, signaling pathway activation, epigenetic regulation, inflammatory microenvironment, and stress response can be just a few of the factors that significantly influence ADSCs differentiation.

## 4. Challenges and Future Perspectives

Although ADSCs remain a highly promising method in regenerative medicine due to their accessibility and the simplicity of isolating them in large quantities, as well as their potential to differentiate into multiple cell lineages, several challenges exist that must be addressed for their future use regarding Angiogenetic Potential in clinical practice translation. One such challenge is the safety and efficiency of materials used for the isolation, culturing, and preservation of ADSCs. For instance, to address this issue, culture media without animal-derived reagents should be established and made readily available for use. Additionally, most of the biomaterials used in the process of utilizing ADSCs in TEare derived from animal resources, which can induce long-term immune reactions in recipients. To comprehensively understand the safety and efficiency of these materials, further in vivo studies must be conducted. Furthermore, there is insufficient information about the mechanisms involved in the proliferation and differentiation of ADSCs into ECs, as well as the different formats of ADSC differentiation into ECs. Therefore, for the clinical applications of these cells, more studies focusing on the different signaling pathways involved in the in vitro and in vivo endothelial differentiation of these cells should be undertaken. Despite the allure of immediate clinical application, we advocate for further experimental research conducted in animal models that closely resemble human diseases. Additionally, because of the restricted capability of ADSCs to differentiate into ECs, cell transplantation might result in a relatively suboptimal therapeutic outcome. Additional clinical trials based on ADSC transplantation are expected in the future.

## 5. Conclusions

ADSCs exhibit promising potential in tissue engineering, demonstrating significant angiogenic capabilities and versatile utility. Their individual characteristics, obtained from readily available adipose tissue, make them an attractive source for applications in regenerative medicine. The angiogenetic potential of ADSCs holds great promise for addressing the vascularization challenges often encountered in tissue engineering. Their capability to differentiate into multiple cell lineages, including adipocytes, osteoblasts, and chondrocytes, expands their utility across diverse tissue types. Furthermore, the minimally invasive nature of AT harvest, ease of isolation, and abundance of adipose-derived cells make them a practical choice for researchers and clinicians. As research in this field progresses, understanding the molecular mechanisms underlying the angiogenetic potential of ADSCs will further help in the elaboration of new research strategies. Ex vivo and in vivo studies are essential to validate and optimize the use of ADSCs in therapeutic approaches.

## Figures and Tables

**Figure 1 ijms-25-02356-f001:**
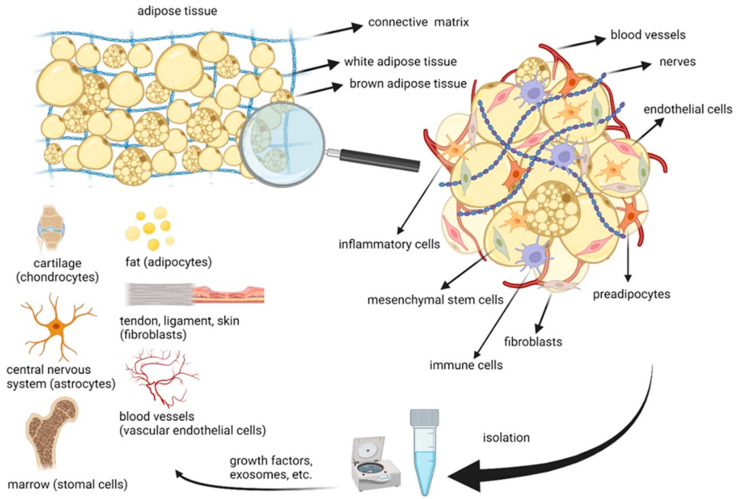
Components and differentiation of ADSCs.

**Table 1 ijms-25-02356-t001:** Functional diversity of AT.

Function	Description/References
Energy storage and metabolic regulation	AT is a central metabolic organ that plays a central role in systemic energy homeostasis. WAT is specialized for the storage of energy (in the form of triacylglycerols) and energy mobilization (as fatty acids), and BAT is a thermogenic tissue. Dysregulation of the regulatory circuits (storage and oxidation in WAT and thermogenesis in BAT) is closely associated with metabolic disorders and AT malfunction, including obesity, insulin resistance, chronic inflammation, mitochondrial dysfunction, and fibrosis [38,39,40,41].
Mechanical protection and body temperature regulation	AT has important mechanical properties, serving to protect organs and cushion body parts exposed to high levels of mechanical stress. Dermal AT is primarily responsible for insulation. This is particularly important in cold environments to prevent heat loss and to help regulate body temperature [42,43].
Endocrine function and hormone storage	The identification and characterization of leptin in 1994 established AT as an endocrine organ. AT was identified as a major site for the metabolism of sex steroids and the production of adipsin. AT secretes a variety of bioactive peptides (adipokines), lipids (lipokines), and exosomal microRNAs, which act at both the local (autocrine/paracrine) and systemic (endocrine) levels. In addition, AT expresses numerous receptors that allow it to respond to afferent signals from traditional hormone systems as well as the central nervous system [44,45].
Appetite regulation	Leptin represents the satiety hormone. The leptin hormone is produced mainly by the gastric mucosa, enterocytes, and adipocytes. This hormone is a marker of energy stores, as triglyceride levels in the fat cells determine the level of leptin secretion [46].

**Table 2 ijms-25-02356-t002:** **ADSCs** utility in tissue engineering.

Study Type	ASCs Sources	Area of Interest	Outcomes	Reference
In vitro	* Human * AT	Hepatic differentiation potential of ADSCs	ASCs can be easily isolated, selected, and induced into mature, transplantable hepatocytes	[175]
In vitro	* Human * AT	ADSCs differentiation into insulin-producing cells	Entiviral vector system could allow the differentiation of ADSCs into insulin-producing cells	[176]
In vitro	* Human * AT	Promote wound healing and improve scaring	Using ADSC-filled sutures can improve wound healing by releasing key molecules involved in angiogenesis, immunomodulation, and tissue remodeling	[177]
Ex vivo	*Human* abdominal fat tissue	Mechanism of ADSCs in wound healing	Topical applications of ADSCs improve wound healing by promoting re-epithelialization and vascularization	[26]
Ex vivo	*Rat* inguinal AT	Mechanisms of ADSCs in preventing allografts immune rejection	Ex vivo infusion of ADSCs prolongs the survival of allografts after surgery	[178]
Ex vivo	*Human* AT from abdomen, buttock, or thigh	Using transduced human ADSCs overexpressing bone morphogenetic protein-2	*Human* ADSCs overexpressing bone morphogenetic protein-2 can heal critical-sized femoral defects	[179]
Ex vivo	* Human * AT	Immunosuppressive and angiogenic activities of ADSCs after coculture with cord blood hematopoietic precursors (cbHSPCs)	ADSCs retain immunosuppressive and proangiogenic capacities with the support of ex vivo expansion of cbHSPCs	[180]
Ex vivo	* Human * AT from the abdomen, buttock, or thigh	Osteogenic potential of cryopreserved ADSCs that are transduced with a *BMP-2*-containing lentiviral vector	ADSCs can be frozen in liquid nitrogen for 3 weeks without any adverse effects to cell viability, protein production, osteogenic potential, or immunophenotype	[181]
Ex vivo	* Human * AT	Potential of ADSCs in fibrosis treatment	Human ADSCs significantly inhibited keloid fibroblast-related bioactivities	[182]
Ex vivo	* Human * AT	Anti-apoptotic and pro-proliferative cytokines secretion of ADSCs’	Direct or indirect contact of ADSCs with ischemic retinal ganglion cells resulted in salvage from cell death	[183]
Ex vivo	* Human * AT	ADSCs’ behavior combined with dermal scaffolds	ADSCs showed a high yield of proliferation and differentiation onto the collagen–elastin matrix of ADSCs	[184]
In vivo (on *rats*)	* Human * adipose tissue	Therapeutic potential of combining ADSCs with modRNA	Significantly improved the retention of fat grafts through proangiogenic and pro-proliferative responses	[185]
In vivo (on *mice*)	Inguinal AT from female *mice*	Role of ADSCs in salivary gland (SG) regeneration	ADSCs-released factors scavenge reactive oxygen species and maintain SG repair and regeneration via paracrine effects	[186]
In vivo (on *rats*)	*Human* AT	Effect of ADSCs in premature ovarian failure	ADSCs transplantation reduced the apoptosis of ovarian granulosa cells and secretion of follicle-stimulating hormone	[187]
In vivo (on *rats*)	*Rat* AT	Differentiation of ADSCs into neural progenitor cells	Study demonstrated the differentiation potential of ADSCs (on fibrin matrix) into transplantable neural progenitors	[188]
In vivo (on *rats*)	*Rat* AT	Action of ADSCs (combined with low-level laser photobiomodulation therapy—LLLT) in the repair process of burned skin	ADSCs+ can improve healing process through significant re-epithelialization, inflammation reduction, and angiogenesis stimulation	[189]
In vivo (on *rats*)	*Human* AT from bariatric surgery	Effect of *human* ADSCs infusion through the cauda equina in rats with traumatic spinal cord injury	This research suggests that immunomodulatory factors secreted by the ADSCs reduced inflammation, inhibited apoptosis, and protected neurons	[190]
In vivo (on *rats*)	*Rat* AT from abdomen	Ability of ADSCs + Resveratrol to promote sciatic nerve regeneration	Application of ADSCs+ could significantly improve the quality of nerve repair compared with untreated ADSCs	[191]
In vivo (in *human*, trial study)	ALLO-ADSCs, approved for clinical studies by the Korean Food and Drug Administration	Safety and efficacy of using allogeneic-ADSCs in the treatment of the anal sphincter of patients with fecal incontinence	Allogeneic-ADSCs have theoretical potential for regeneration of the anal sphincter	[192]
In vivo (in *human*, case report)	*Human* abdomen fat	Effect of ADSCs injection + core decompression in early-stage of avascular necrosis of the femoral head	3 months post op. MRI showed healed femoral head with a recession of the lesion	[193]
In vivo (in *human*, trial study)	Autologous ADSCs obtained by lipoaspiration from abdominal subcutaneous fat	Efficacy and safety of a single intra-articular injection of ADSCs for patients with knee osteoarthritis	No changes in MRI of cartilage defect at 6 months vs defect increase in the control group. ADSCs helps to functional improvement and pain relief for patients with knee osteoarthritis, without causing adverse events at 6 months follow-up	[194]
In vivo (in *human*, trial study)	Autologous MSCs obtained by liposuction from the inner face of the thighs	To evaluate the effects of cell therapy with ADSCs on the treatment of detrusor underactivity in men	ADSCs therapy led to improvements in voiding function	[195]
In vivo (in *human*, trial study)	Allogeneic ADSCs from healthy donors obtained by liposuction from abdominal subcutaneous AT	To investigate if a single treatment with direct intramyocardial injections of ADSCs was safe and improved cardiac function in patients with chronic ischemic heart failure with reduced ejection fraction.	Direct injection with allogeneic ADSCs into the myocardium was safe during a 3-year follow-up period. However, in comparison to placebo, there was no significant improvement of left ventricular volumes or function, or clinical symptoms 6 months after treatment	[196]
In vivo (in *human*, trial study)	Autologous ADSCs obtained by liposuction from subcutaneous AT	Evaluation of the periodontal defects regeneration using a mixture of ADSCs and PRP (platelet-rich plasma)	Cell therapy using ADSCs can represent a useful medical technology for regeneration of periodontal defects	[197]

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
