# Peer review of "Adipose-Derived Stem Cells: Angiogenetic Potential and Utility in Tissue Engineering"

_ijms, 2024, doi:10.3390/ijms25042356_

Round 1

Reviewer 1 Report

Comments and Suggestions for Authors

The manuscript by Biniazan and colleagues reviews the angiogenetic potential and application of adipose-derived stem cells in tissue engineering. Overall, while the manuscript is relatively straightforward, I think the review is simplistic and generally does not provide new information. Based on this consideration, a major revision is warranted in order for the manuscript to be considered for publication. Please find the detailed comments/suggestions below:

1.     Firstly, please clarify what is new from this review compared to previously published review, such as https://doi.org/10.1155/2016/6737345 or https://doi.org/10.3390/ijms221910890.

2.     Secondly, considering the main content is regarding ASCs, “Section 2. Adipose Tissue Function and its Component” is not within the scope of the review. There are plenty of possible discussion points that can be incorporated rather than that section. Especially considering that angiogenesis is the main point (mentioned in the title), the current content in subsection “2.1. (should be 3.1.) Impact on Angiogenesis” was too short. A more intricate explanation should be provided.

3.     The differentiation of MSCs, including ASCs, beyond their “supposed” lineage (adipogenic, chondrogenic, or osteogenic differentiation) that the author mentioned as “multiple cell lineage” (into the neurogenic lineage, for example) is reported to not occur in all populations but only occur in a small sub-population of the MSCs called Muse cells (see: https://doi.org/10.1089/scd.2013.0473,https://doi.org/10.1371/journal.pone.0064752). Information regarding this point is important to be discussed and highlighted.

4.     One of the main downsides of ASCs is the lower “quality” of the stem cells compared to those isolated from fetal tissue (e.g., proliferative capacity, differentiation capacity, and limitation in passaging). In clinical cases, it is quite well-known that ASCs quality depends on the donor, where unhealthy donors will generate lower “quality” stem cells. This issue should be addressed in this review.

5.     The important part of a review article is the critical analysis by the authors. The main body shows the current state-of-the-art. Based on this knowledge, a longer section should be added to address what is missing from the current knowledge and what should be the main concern or direction of research in the future.

6.     The current manuscript still contains numerous errors. For example:

·      Consistency in using ASCs vs ASC vs ASCS throughout the manuscript.

·      Double space in Line 14, 17, 20

·      Line in Table 2 is not organized.

·      Reference in Table 2 can be simplified into just citation number.

·      The subsection in Section 3 is currently written as 2.1, 2.2, and so on. It should be written as 3.1, 3.2, and so on.

·      The acknowledgment statement is too large. The authors should acknowledge key people who assist in the writing or literature search, preferably by their individual name or institution (e.g., member of xxx laboratory).

Author Response

Thank you for your time and all the comments.

Reviewer 2 Report

Comments and Suggestions for Authors

The present manuscript review the use of adipose derived stem cells in TE, with special attention on their angiogenic potential. The scope of the review fits the journal and is of interest to researchers in the area but, while it is well written and structured, this reviewer thinks that could be completed if some topics are discussed and included in the review.

11)     A brief description of angiogenesis (a figure would be helpful) should be included, with special attention to role players (cells), growth factors, cytokines and their evolution over the time of healing. While this is very briefly discussed in other section of the review, it requires a further description given the topic of the review.

22)     Overall in the whole review, but with special emphasis on section 2.1., this reviewer has the feeling that the review would greatly improve with a deeper description at mechanistic levels of the pathways involved in angiogenesis and the role that ASCs (or derived/secreted factors) play on it. Section 2.1. should be expanded and include the molecular pathways involved (at least in the major growth factors such as VEGF), include examples of relevant cytokines, which pathways microRNAs are interfering with… etc. This is a field with a plethora of studies in the field.

33)     Section 2.2. How relevant is differentiation of ASC into endothelial cells, given the relatively low number compared to host’s cells when applying therapy? Or do the authors refer to the events happening in vivo in normal conditions? This is not very clear in this section and could be addressed with further discussion.

44)     In section 2.4. Further expansion on mechanistic issues at molecular level should be discussed, or maybe summarized after table 2 if these are common to most of areas of research. For instance, which factors are the main active role players in each TE, VEGF? In the case of immunomodulation studies, which phenotypes or on which cell types (myeloid, Tcells) and which secreted cytokines are being involved? Also, vascular and cardiac should be further discussed.

55)     A section that explains and includes studies of different formats of angiogenesis induced by ASCs should be included (also a table could reduce text writing). Throughout the review, different formats are named: cell therapy, secretome, exosomes, tissue engineered materials + cells, combination with drugs, etc.; but not discussed. Which formats have shown higher induction of angiogenesis and in which applications? Which are the drawbacks and advantages of each one?

Minor comments:

66)     Is it not the term ADSC more common than ASC? This is a minor note, but should be considered.

77)     Line 82, the term SVF is included but not specified previously (it is later).

88)     Line 103. Is not sex of the donor a determining factor of yield / characteristics?

99)     Line 169. What does “poor conditions” refer to? This term is too vague

110)  Line 179. Does Omission of FGFb implies blocking? Knock out?

111)  Table 2. What is the difference between ex-vivo / in vitro? Are ex-vivo referred to harvested tissue models? The studies / rows should be shorted in order (in vitro / ex vivo / in vivo). Also, in vivo in humans should be classified as Clinical trial / case series. Also, more clinical trials studies should be included. Finally, the title of the reference column can be removed.

Comments on the Quality of English Language

Minor typos detected. General review required

Author Response

(The authors gave the same response as above.)

Reviewer 3 Report

Comments and Suggestions for Authors

This review highlights the topic of adipose-derived stem cells from the perspective of their potential for angiogenesis and tissue engineering. The review is easy to read and well structured. The chosen topic is not original, so it would be great to see a list of the advantages of your work over other reviews on the topic ‘adipose-derived stem cells’. After the authors will take into account all the comments recommendations, the manuscript can be published.

The comments include requirements and recommendations.

1) ‘Several concerns have been raised regarding the procurement human embryonic stem cells [19]’ (lines 44-45): It is not clear how this reference relates to the statement. Perhaps if you list the concerns it will be clearer. The sentence doesn't make enough sense.

2) Despite the specifics of the journal, it is better to decipher the following abbreviations: ‘FGF, HGF, PDGF’ (line 148), ‘VEGF’ (line 154), ‘Ac-LDL’ (line 179), ‘vWF’ (line 185), ‘ERK’ (line 201).

3) ‘The purified ECs from both adipose and dermal origins exhibited comparable expression levels of standard endothelial surface markers, including PECAM-1, VE-Cadherin, and VEGFR2 [81]’ (lines 165-167): The thesis does not match the reference. In the article you link to there is nothing about endothelial cells from adipose and dermal origins. Please reference appropriate article.

4) ‘Santerre et al.’ (line 202): Usually the first author is indicated. I would recommend to place the reference [83] at the first mention (line 25).

5) References (line 331):

Add doi where it is available, for example, you can get ‘doi: 10.2174/138161211795164220’ to ‘Gimble JM, Nuttall ME. Adipose-derived stromal/stem cells (ASC) in regenerative medicine: pharmaceutical applications. Curr Pharm Des. 2011;17(4):332–9’ (lines 383-384), etc..

There are no references numbered over 155 in the text though there are references [155-164] mentioned in the ‘Reference’ section (lines 664-686). Please correct this issue.

Author Response

Thank you very much for your time and all the comments.

Round 2

Reviewer 1 Report

Comments and Suggestions for Authors

The authors have addressed the comments and suggestions. The manuscript can be considered for publication.

Author Response

Thank you one more time for all the suggestions and comments. 

Reviewer 2 Report

Comments and Suggestions for Authors

The review has been greatly improved and the information provided is very well organized, more complete and useful to the reader. Authors have addressed all my comments.

1) Only a minor note on the description of the angiogenesis. (Line 165): This description seems very specific on the tracheal environment, and should be generalized. I suggest papers like Eelen et al. (2020) - https://doi.org/10.1161/CIRCRESAHA.120.316851 - could be use as resource for summarizing angiogenesis in a general way. 

Minor comments: 

2) FGF basic is described as FGF-2 in some parts of the text and as FGFb in others. Please revise. 

3) In the table, ADSC still read as ACS

4) line 412, "artery Ecs" should read "ECs"

5) Abbreviations: ASCs should be updated to ADSCs

Comments on the Quality of English Language

Minor typos should be revised

Author Response

Thank you very much for all the comments. 
All suggestions were addressed. 
Line 165 (description of the angiogenesis) was changed to more general info and the suggested reference was added.